# The Prognostic Role of the Left Atrium in Hypertensive Patients with HFpEF: Does Function Matter More than Structure?

**DOI:** 10.3390/life15091483

**Published:** 2025-09-21

**Authors:** Artem Ovchinnikov, Alexandra Potekhina, Anastasiia Filatova, Olga Svirida, Maria Sobolevskaya, Alfiya Safiullina, Fail Ageev

**Affiliations:** 1Laboratory of Myocardial Fibrosis and Heart Failure with Preserved Ejection Fraction, National Medical Research Center of Cardiology Named After Academician E.I. Chazov, Academician Chazov St., 15a, 121552 Moscow, Russia; potehina@gmail.com (A.P.); anastasia.m088@yandex.ru (A.F.); olgasvirida@yandex.ru (O.S.); msobolevskaya95@mail.ru (M.S.); 2Department of Clinical Functional Diagnostics, Russian University of Medicine of the Ministry of Health of the Russian Federation, Dolgorukovskaya St., 4, 127006 Moscow, Russia; 3Laboratory for Improving Medical Care for Patients with Coronary Heart Disease, National Medical Research Center of Cardiology Named After Academician E.I. Chazov, Academician Chazov St., 15a, 121552 Moscow, Russia; 4Laboratory of Cell Immunology, National Medical Research Center of Cardiology Named After Academician E.I. Chazov, Academician Chazov St., 15a, 121552 Moscow, Russia; 5Department of Myocardial Diseases and Heart Failure, National Medical Research Center of Cardiology Named After Academician E.I. Chazov, Academician Chazov St., 15a, 121552 Moscow, Russia; a_safiulina@mail.ru; 6Out-Patient Department, National Medical Research Center of Cardiology Named After Academician E.I. Chazov, Academician Chazov St., 15a, 121552 Moscow, Russia

**Keywords:** left atrium, heart failure with preserved ejection fraction, strain, diastolic dysfunction, outcome

## Abstract

Background: In arterial hypertension (AH), adverse hemodynamic consequences in the left atrium (LA) are often observed. The prognostic significance of functional vs. structural LA abnormalities among high-risk AH patients (with heart failure with preserved ejection fraction [HFpEF]) are not clearly defined. Objective: to compare the prognostic significance of structural vs. functional LA indices in hypertensive patients with HFpEF. Methods: We retrospectively selected 274 hypertensive patients with AH, HFpEF, and sinus rhythm. The primary outcome was a composite of all-cause mortality and HF hospitalization; the median follow-up was 4.3 (2.5–6.5) years. Results: The composite endpoint occurred in 133 patients (49%). Kaplan–Meier analysis revealed significantly lower event-free survival rates in patients with lower functional LA reservoir strain [LASr] (≤median) compared to patients with higher LASr (*p* < 0.001). Patients with higher structural LA volume index (LAVI) as well as with higher LV filling pressure (E/e′ ratio) or more severe left ventricular (LV) hypertrophy (higher LV mass index) had a similar prognosis to patients with lower values. In multivariable analysis, decreased LASr and paroxysmal atrial fibrillation (AF) were independently associated with adverse outcomes after accounting for potential confounders (for both *p* < 0.05). Conclusions: Among patients with AH and HFpEF, the functional LA parameter LASr seems to be more effective than the structural LA parameter LAVI, or traditional indexes of LV hypertrophy and filling pressure, in predicting prognosis.

## 1. Introduction

Arterial hypertension (AH) is a major risk factor for the development of HF, mainly HF with preserved ejection fraction (HFpEF). As a consequence, the great majority of patients with HFpEF (>90%) have a history of AH [1]. AH is characterized by the development of concentric left ventricular (LV) hypertrophy with progressive LV diastolic dysfunction (LVDD) and increased LV filling pressures [2]. Although diastolic abnormalities are the most common cardiac complications of AH and play a crucial role in the subsequent development of HFpEF, the pathophysiology of hypertensive heart disease has been found to be associated with adverse structural and functional changes in the left atrium (LA) [3]. Pressure overload of the LV results in higher LV filling pressures, which, in turn, can increase pressure in the LA and trigger atrial remodeling. AH may also affect the LA via neurohumoral mechanisms [4].

Left atrial abnormalities appear to be a pivotal event in the natural course of hypertensive heart disease and HFpEF [5]. Normally, the LA functions as a buffer, protecting the vulnerable pulmonary microvasculature and right heart chambers from the damaging effects of increased LV end-diastolic pressure (EDP) [6]. However, as soon as LA impairments occur, high LV EDP will be transmitted to the pulmonary circulation leading to pulmonary venous congestion [7], worsening lung function (increased ventilatory drive and impaired pulmonary ventilation reserve) [8], and the development of PH with right ventricular (RV) failure [6,9,10]. Hypertension-mediated LA remodeling also augments the risk of developing atrial fibrillation (AF), which has negative clinical and prognostic consequences [11].

Hypertension-mediated LA remodeling is characterized by LA enlargement and dysfunction, which can be reliably assessed using echocardiography. Structure is typically evaluated using LA volume indexed to body surface area (LAVI), while function is assessed using LA strain measures, primarily reservoir strain (LASr), via speckle tracking echocardiography [12]. Both structural and functional indicators of the LA exhibits diagnostic and prognostic significance in HFpEF [12,13,14] and widely used as a surrogate measure of LV diastolic function/filling pressure [15,16]. Nevertheless, LA dilatation may fail to detect the early stages of HFpEF, when the mean LA pressure only rises under stress, and the LA requires time to remodel. On the other hand, LA functional dysfunction is highly sensitive to the initial cardiac impairments and becomes evident at the earliest stages of HFpEF [13,17]. Thus, LA dysfunction may be a more robust prognostic predictor than structural parameters in patients with HFpEF. The aim of this study was therefore to evaluate the long-term prognostic significance of structural vs. functional LA parameters and to compare the prognostic impact of LA parameters to that of parameters reflecting LV diastolic function/filling pressure or LV hypertrophy in hypertensive patients with HFpEF.

## 2. Materials and Methods

### 2.1. Study Population

The study included consecutive patients aged ≥ 40 years with HFpEF who visited the Outpatient Department of the National Medical Research Centre for Cardiology in Moscow, Russian Federation, between January 2015 and November 2022. Basic clinical data for the baseline examination including clinical status, concomitant diseases, electrocardiographic, echocardiographic, and radiographic data were collected from archival documents. Study participants had symptoms and signs of HF, preserved LV ejection fraction (≥50%), and elevated LV filling pressure that was verified at rest or during exercise via echocardiography [15]. Elevated LV filling pressure at rest was verified if grade II-III LV DD was detected, and that during exercise if exercise-induced elevations in average E/e′ ratio > 14 and TR velocity > 2.8 m/s were observed [15]. We excluded patients with permanent atrial fibrillation (AF) because this condition has a significant independent influence on LA structural and functional parameters. Other exclusion criteria were the evidence of myocardial ischemia during stress echocardiography or significant untreated stenoses of epicardial coronary arteries, LV dilatation (LV end-diastolic dimension ≥ 5.9 cm in men and ≥5.3 cm in women), inability to complete exercise or inadequate acoustic windows, artificial cardiac rhythm or left bundle branch block, significant structural valve disease, significant mitral annulus calcification, diseases associated with isolated right ventricular failure, hypertrophic cardiomyopathy, infiltrative or inflammatory myocardial diseases, and pericardial diseases.

The primary outcome was time to death from any cause or to the first hospitalization for exacerbation of HF during follow-up. Follow-up clinical data and required prognostic information were collected retrospectively from baseline through electronic medical records.

Death and its cause were confirmed and clarified by the submission of a request to the Russian social and medical services. Hospitalization for exacerbation of HF was defined as meeting the following criteria: (1) HF as the primary diagnosis at admission; (2) confirmation of new or worsening symptoms and/or signs of HF at admission; and (3) intravenous administration of loop diuretics during hospital stay. The study was approved by the Independent Ethics Committee of the National Medical Research Center of Cardiology (Protocol No. 273) and complied with the Declaration of Helsinki. The trial was registered at ClinicalTrials.gov Identifier: NCT06844032.

### 2.2. Study Design

Of 352 consecutive patients with a diagnosis of HFpEF and sinus rhythm, 78 patients were excluded due to various exclusion criteria. Thus, the final cohort consisted of 274 patients with HFpEF for whom comprehensive prognostic information had been obtained (Figure 1). The median follow-up period was 4.3 (2.5–6.5) years. During the follow-up period, 48 patients (18%) died, and 85 patients (31%) were hospitalized for worsening HF. Thus, the primary composite endpoint, incorporating both all-cause mortality and hospitalization for HF exacerbation, occurred in 133 patients (49%). The remaining 141 patients (51%) constituted the control group, comprising survivors without hospitalization for HF exacerbation.

The meticulous clinical evaluation including the assessment of the New York Heart Association (NYHA) functional class and 6 min walk test distance (6MWD), echocardiography (at rest and during bicycle supine exercise, or DST), and blood analyses for the N-terminal pro-brain natriuretic peptide (NT-proBNP), biomarker of myocardial stress were performed at the time of the patient’s visit to the Outpatient Department (baseline visit), and the follow-up started at this point.

### 2.3. Echocardiography

The echocardiography was performed on Vivid S70 and Vivid E95 (GE Healthcare, Horton, Norway) ultrasound machines by experienced cardiac sonographers. LV wall thickness, chamber dimensions and volumes, mass, ejection fraction, and LA maximal volume, were determined according to the current guidelines [18]. The LA maximal volume and LV mass were indexed to body surface area (LAVI and LVMI, respectively). LA dilatation was considered if the LA maximal volume index (LAVI) was greater than 34 mL/m^2^. LV hypertrophy was defined as LV mass index > 115 g/m^2^ in men and >95 g/m^2^ in women. The relative wall thickness (RWT) was defined as [interventricular septal thickness + posterior wall thickness]/LV end-diastolic dimension, with further classification of increased LV mass index as concentric (RWT > 0.42) or eccentric (RWT ≤ 0.42) hypertrophy; in the case of normal LV mass and RWT > 0.42, concentric remodeling was diagnosed [18].

LV diastolic function was assessed by measuring pulsed Doppler mitral peak early diastolic (E-wave) and late diastole (A-wave) flow velocities and their ratio (E/A); average tissue Doppler-derived mitral annulus relaxation velocity (e′), the mitral E/e′ ratio, the left atrial maximum volume index (LAVI), and the tricuspid regurgitation (TR) velocity via continuous-wave Doppler [15]. The severity of LVDD was determined according to the 2016 ASE criteria for the classification of LV diastolic dysfunction [15]. Briefly, an E/A ratio of 0.8 or less with a peak E-wave velocity of 50 cm/s or less indicates grade I LVDD, while an E/A ratio of 2 or more suggests grade III LVDD. For patients between these two extremes, determine how many of the LV hypertension criteria are fulfilled: E/e′ > 14, LAVI > 34 mL/m^2^, and peak TR velocity > 2.8 m/s. If two or more criteria are met, then grade II LVDD is present; if two or more criteria are not met, then grade I LVDD is present [15].

The assessment of right ventricular (RV) systolic function included M-mode tricuspid annular plane systolic excursion (TAPSE). Normally, TAPSE exceeds 1.7 cm [18]. Pulmonary artery systolic pressure (PASP) was calculated as a sum of the peak TR jet velocity and right atrial (RA) pressure on the basis of inferior vena cava’s (IVC) size and its collapse [18]. The RA pressure was considered equal to 3 mm Hg if the collapse of the IVC after a deep breath exceeded 50% and its diameter was no more than 21 mm. If the diameter of the inferior vena cava exceeded 21 mm and it did not decrease by more than 50% after a deep breath, the RA pressure was considered to be equal to 15 mm Hg. In intermediate cases, RA pressure was considered equal to 8 mmHg. Pulmonary hypertension was diagnosed when the estimated PASP was above 35 mm Hg [18].

### 2.4. Speckle Tracking Analysis

The deformation analyses of left heart strain variables via 2-dimensional speckle tracking echocardiography (LV global longitudinal strain [GLS] and LA global longitudinal reservoir strain [LASr]) were performed in all patients. Relevant images and cine loops were stored as DICOM (Digital Imaging and Communications in Medicine) files on a server and analyzed offline using the dedicated ultrasound software package (Echo-Pac version 203, GE Healthcare) at frame rates of 50–80 frames/s. Ventricular end-diastole was determined as the time reference to define the zero baseline for LA strain curves; LASr was calculated as the average strain in six segments of the LA in a nonforeshortened apical four-chamber view [19]. GLS was measured as the average of the systolic strain obtained from all the LV segments in the apical 4- and 2-chamber and long-axis views. An abnormal LASr was defined as <23% [20] and GLS as <16% [21]. We also calculated the LA stiffness as the ratio of E/eʹ to LASr (Figure 2) [22]. All echocardiographic measures represent the mean of ≥3 beats.

To determine the reproducibility of LA parameters, we analyzed intra- and inter-observer variability in 20 randomly selected patients using intraclass correlation coefficients. Reproducibility was adequate for both LASr (intraclass correlation for intra- and inter-observer variability 0.97 [95% CI 0.93 to 0.98] and 0.91 [95% CI 0.83 to 0.96], respectively) and LAVI (intraclass correlation for intra- and inter-observer variability 0.95 [95% CI 0.89 to 0.97] and 0.90 [95% CI 0.76 to 0.96], respectively).

### 2.5. Diastolic Stress Test (DST)

The DST was mainly performed to diagnose heart failure (HF) in patients with a high probability of having of HFpEF but normal LV filling pressure at rest (i.e., grade I LVDD), as well as to exclude myocardial ischemia or to assess the load tolerance. All these reasons were independent of current study’s purposes. Patients exercised supine bicycle ergometry with a pedaling frequency of 60 rpm starting with a 3 min period of a low-level 25 Watts (W) workload followed by 25 W increments in 3 min stages to the maximal tolerated levels or until the patient developed limiting symptoms. During the test, the changes in LV filling pressures (the mitral E/e′ ratio and TR velocity) were analyzed. These parameters were recorded initially, further at each stage of the load, and then at the peak load. We used the average value of multiple beats (≥5 consecutive heart cycles) to minimize measurement error due to exaggerated respiration variation during exercise test. Elevated LV filling pressure at exercise was verified if exercise-induced elevations in average E/e′ > 14 and TR velocity > 2.8 m/s during DST were observed [15].

### 2.6. NT-proBNP

Blood samples were taken into tubes with citrate anticoagulant (to obtain plasma) by venous puncture after a 20 min supine resting period. The obtained blood samples were immediately centrifuged to separate plasma, which were then frozen and stored for a maximum of 6 months at −80 °C. Blood samples were not allowed to be thawed and refrozen. Levels of the N-terminal pro-brain natriuretic peptide (NT-proBNP) biomarker of myocardial stress in the plasma were measured via an automated electrochemiluminescence immunoassay (Roche Diagnostics, Mannheim, Germany).

### 2.7. Statistical Analysis

Statistical analysis was performed via standard software (MedCalc, version 19.5.3, Ostend, Belgium). Normally distributed data are presented as the means ± standard deviations; nonnormally distributed data are presented as medians and interquartile ranges (IQRs). Categorical variables are reported as numbers and percentages of observations. The differences in quantitative parameters between patients who died or were hospitalized due to HF with those who did not experience such complications were tested using Student’s t test for normally distributed variables, the Mann–Whitney U test for nonnormally distributed variables, and the χ^2^ or Fisher’s exact test as appropriate for qualitative data. To assess the diagnostic accuracy of a certain variable in identifying patients with an adverse prognosis, ROC analysis was conducted. Spearman’s rank correlation method was used to determine the correlation between the variables. A receiver operating characteristic (ROC) analysis was performed to assess the predictive performance of LA variables and variables of LV diastolic dysfunction/filling pressure for the primary endpoint (the composite outcome of all-cause mortality and HF hospitalization). The best cutoff value of variables was derived from the ROC curve as the value with the highest sum of sensitivity and specificity. The areas under the curve (AUC) were compared using the DeLong test.

Survival rates were estimated using the Kaplan–Meier analysis, and the significance level was assessed by the log-rank test. A Cox proportional hazards multivariate regression model was used to identify independent predictors of adverse outcome. Candidate parameters were selected among those that were either clinically relevant or statistically significant in the univariate analysis. To eliminate multicollinearity, the selected variables were analyzed for correlations with each other. Correlation analysis was performed using Pearson’s criterion for continuous characteristics (φ coefficient for dichotomously distributed characteristics). Variables that were cross-correlated and had a weaker correlation with the outcome variable were excluded. Three different models were subsequently constructed: model 1—non-adjusted; model 2—adjusted for common clinical parameters (age, sex, and BMI); and model 3 adjusted for the same clinical parameters as in model 2 as well as for the severity of AH (LV mass index and systolic BP), a biomarker of HF severity (blood NT-proBNP level), and comorbidities relevant for HFpEF such as chronic kidney disease (CKD) and type 2 diabetes mellitus (T2DM).

The value of *p* < 0.05 was considered statistically significant.

## 3. Results

### 3.1. Patient Baseline Characteristics

The study participants represented a typical outpatient cohort of HFpEF (Table 1): predominantly elderly patients (median age 68.7 years) with moderate functional limitations (58% had NYHA functional class II and 27% had functional class III), mild/moderate elevation of NT-proBNP (median 283 pg/mL), and multiple comorbidities (91% had overweight or obesity, 45% had paroxysmal AF, 41% had CKD, 40% had T2DM, and 35% had ischaemic heart disease). At the time of the initial visit, more than half of the study participants (61%) had evidence of increased LV filling pressure at rest (grade II–III LVDD), indicating advanced stages of HFpEF. This assertion is further substantiated by the observation that 84% of patients exhibited LA enlargement, 69% had LV hypertrophy, and 50% had pulmonary hypertension (Table 2).

Most patients were taking renin–angiotensin system blockers (ACE inhibitors or angiotensin receptor blockers); more than half were taking beta-blockers, statins, and loop diuretics. Given that most patients were enrolled in studies before the era of modern HFpEF therapy, only a small number were taking mineralocorticoid receptor antagonists, angiotensin receptor–neprilysin inhibitors (ARNI) and sodium–glucose cotransporter 2 (SGLT2) inhibitors (Table 1).

There was a gradual decrease in LASr and LA stiffness and increase in LAVI levels from I through II to III LVDD grades. In addition, patients with paroxysmal AF had higher maximum LA volumes and lower LA functional indices (LASr and LA stiffness) compared with patients without AF (Figure 3a–c). LA functional parameter showed good correlations with structural (LAVI) one (Figure 3d,e).

### 3.2. Comparison of Patients Who Died or Were Hospitalized Due to HF Exacerbation with Event-Free Survival Patients

Multiple differences were found when comparing the baseline characteristics of patients who died or were hospitalized for HF (*n* = 133) with those without these complications (*n* = 141) (Table 1 and Table 2). The died or HF hospitalized patients were older and more likely to have paroxysmal AF. They experienced more severe HF: they took loop diuretics more often and had mild functional limitations (NYHA functional class I) less often. They had moderate-to-severe LVDD (grade II or III) more often, which is confirmed by higher LAVI, PASP, LV mass index, and NT-proBNP level in blood. These patients also exhibited poorer LA functional status, as evidenced by lower LA strain during the reservoir phase (LASr) and greater LA stiffness (the E/e′ to LASr ratio, Table 2).

Patients who died or were hospitalized due to HF had significantly higher PASP but did not differ in E/e′ ratio, a key parameter of LV filling pressure, compared with event-free survival patients. There was no difference in LV (ejection fraction and GLS) and RV (TAPSE) contractility between the compared groups (Table 2).

### 3.3. The Predictive Performance of Echocardiographic Parameters of LA Structure/Function and Parameters of LV Diastolic Dysfunction

We compared prognostic performance of echocardiographic LA structural (LAVI) and LA functional (LASr, and LA stiffness) parameters with those reflecting LV diastolic dysfunction/filling pressure (E/e′ ratio, TR velocity, and LVMI) for predicting unfavorable outcomes (Figure 4). The prognostic performance of LASr was highest (cutoff value 20%) with an AUC 0.67 (95% CI 0.61 to 0.72), *p* < 0.0001, and was better than structural LA parameter LAVI (*p* = 0.050 for differences in AUCs) and parameters reflecting LV filling pressure (E/e′ ratio and PASP; *p* < 0.05 for differences in AUCs, Figure 4). Interestingly, a LASr of less than 18%, which is consistent with an elevated LAP [23], had a very high specificity of 92% (with a sensitivity of 35%) for identifying patients with poor outcomes. LA reservoir strain was also superior to LVMI—a parameter reflecting LV hypertrophy and indirectly LVDD (*p* = 0.04 for difference in AUCs; Figure 4).

Similarly to LASr, LA stiffness had significantly higher predictive performance than the E/e′ ratio and PASP. The structural LA indicator LAVI did not differ from these two key markers of LA filling pressure in terms of predictive significance (Figure 4).

### 3.4. Identification of Prognostic Markers for HFpEF

The Kaplan–Meier method was used to assess clinical outcome rates in patients with different structural and functional LA characteristics. Analysis revealed significantly lower event-free survival rates in patients with lower LASr (≤median) compared to patients with higher LASr (>median), *p* = 0.0002 (Figure 5a). Similarly, patients with higher LA stiffness (>median) experienced a significantly worse prognosis compared with those with lower LA stiffness (≤median) (*p* = 0.004, Figure 5b).

Patients with paroxysmal AF experienced more adverse events than patients without AF (*p* < 0.0001; Figure 5c). At the same time, patients with higher LAVI (Figure 5d), as well as those with higher E/e′ ratio, PASP or LVMI had a similar prognosis to patients with lower values of these parameters (for all comparisons *p* > 0.05; Figure 5e–g).

Among the available clinical and echocardiographic variables, paroxysmal AF, structural and functional LA parameters (LASr, LA stiffness, and LAVI), use of loop diuretic, NYHA functional class, and LVDD grade were found to be statistically significant univariate prognostic covariates (Table 3). To select candidates for Cox’s proportional hazards model, these as well as clinically relevant parameters (age, sex, BMI, systolic BP, NT-proBNP, LV mass index, CKD, and T2DM) were analyzed to eliminate multicollinearity. Based on collinearity (r > 0.5 or <−0.5), LA stiffness was excluded from the final Cox model due to a strong negative correlation with LASr (r = −0.72 [95% CI −0.77 to −0.65], *p* < 0.0001) and as having a less significant association with the outcome (Table 3).

A multivariate analysis using a Cox proportional hazards model revealed that LASr and paroxysmal AF were significant and independent predictors of clinical events across various models: model 1 (non-adjusted)—hazard ratio (HR) 0.95 (95% CI 0.92–0.99), *p* = 0.023 and HR 1.70 (95% CI 1.16–2.48), *p* = 0.006, respectively; model 2 (adjusted for age, sex, and BMI)—HR 0.95 (95% CI 0.91–0.99), *p* = 0.017 and HR 1.70 (95% CI 1.16–2.49), *p* = 0.007, respectively; and model 3 (adjusted for age, sex, BMI, NT-proBNP, LV mass index, systolic BP, CKD, and T2DM): HR 0.95 (95% CI 0.90–0.99), *p* = 0.012 and HR 1.76 (95% CI 1.18–2.62 (*p* = 0.005), respectively (Table 3).

## 4. Discussion

This retrospective cohort study compared the prognostic significance of echocardiographic structural LA with functional LA parameters, as well as with parameters of LV filling pressure in a well-characterized outpatient cohort of hypertensive patients with HFpEF. The results of this study suggest that LA functional parameters may be more closely associated with an increased risk of death or HF hospitalization than LA structural/anatomical or LV filling pressure indices. First, the prognostic performance of functional LA reservoir strain was higher than that structural LA parameter LAVI, as well as parameters of LVDD/filling pressure (E/e′ ratio and PASP) or LVH severity (LV mass index). Second, a lower LA reservoir strain, along with paroxysmal AF, were powerful and independent predictors of an adverse long-term outcomes in hypertensive patients with HFpEF, even after adjusting for clinically relevant parameters and indexes of the AH and HF severity. In contrast, neither LAVI nor measurements of LVDD/filling pressure or LV hypertrophy predicted clinical outcomes in the multivariate analysis. The observed prognostic benefit of LA function appears particularly significant in the context of AH, given the traditional emphasis placed on LV hypertrophy and diastolic dysfunction in the pathogenesis of hypertensive HF.

Similarly to other studies, this study showed that both structural and functional LA parameters deteriorate as LVDD severity increases [24,25]. However, there are fundamental differences between these two types of LA impairments. The LA, being a thin-walled structure with limited capacity for compensatory hypertrophy, cannot withstand sustained pressure overload for a long time, and as soon as LA mean pressure becomes permanently elevated, the LA cavity begins to dilate [5]. Indeed, LA enlargement is a reliable and well-established indicator of increased LV filling pressure [13] and an integral part of current guidelines for assessing LV diastolic function/filling pressure [15,16]. Nevertheless, LA dilatation is a relatively late event in the course of in HFpEF and may not be present in the early stages, when LV filling pressure increases only during exercise. By contrast, LA dysfunction can be identified from the earliest stages of HFpEF [17,26] with progressively worsening with increasing disease severity [25]. Thus, echocardiographic markers of LA functional abnormalities could more closely ‘accompany’ the natural course of HFpEF compared to LA volume or parameters of LV diastolic function [13]. In the present study, a significant proportion of patients (39%) had early-stage HFpEF (i.e., LVDD grade I) with normal or mildly dilated LA, which may partially explain the observed prognostic discrepancies between LA structural and functional parameters.

These differences can also be explained by cause-and-effect interdependence, because LA dysfunction is the main cause of increased mean LA pressure and its subsequent dilatation. In subjects with isolated LV relaxation (grade I LVDD), LV filling depends mainly on enhanced contraction of the LA (known as the “atrial booster pump”), which ensures adequate ventricular filling at normal mean LA pressure. However, as LVDD progresses and the LV becomes stiffer, atrial booster pump may weaken. As a result, a ‘compensatory’ passive increase in mean LA pressure occurs. This will help keep the noncompliant LV filled up but cause or worsen HF symptoms [5].

The LA modulates ventricular filling through three intrinsic functions: reservoir, conduit, and contractile. These functions can be measured accurately using 2D speckle tracking echocardiography [19]. In the present study, LA function was assessed by its distensibility in the reservoir phase (LASr). To date, LASr is the most common and well-studied LA mechanics parameter [27]. Measurement of LASr has excellent clinical feasibility and applicability [23], which was confirmed in the present study. Normally, LA reservoir function ensures adequate atrial filling at low pressures. LASr is determined by the properties of the LA, such as its distensibility, relaxation and contractility, as well as contractility of both ventricles [28]. The dependence of LA reservoir function on multiple factors makes it sensitive to the earliest cardiac impairments [17] with gradual deterioration as HFpEF progresses [25]. In patients with HFpEF, LASr moderately correlates with invasively measured LV filling pressure [23,29] and is more accurate than conventional diastolic parameters in revealing of HFpEF [30]. A meta-analysis of Khan FH et al. has shown that decreased LASr has prognostic value in HFpEF patients [14].

In the present study, we also used LA stiffness (E/e′ ratio to LASr) to assess atrial function. Like LASr, atrial stiffness had significantly higher predictive performance than the conventional measures of LV diastolic function/filling pressure, and increased LA stiffness was associated with increased risk of adverse outcomes. These results are consistent with other studies [22], although LA stiffness may be more valuable for diagnosing HFpEF than LA reservoir dysfunction [30]. Thus, our results suggest that both LASr and LA stiffness could be used as simple and reliable imaging biomarkers to indicate the severity and prognosis of HFpEF.

Unlike LA function indices, key echocardiographic parameters of LVDD such as E/e′ ratio, PASP, or parameters of LV hypertrophy such as LVMI were not associated with adverse outcomes. Similar results were obtained in a study by Freed BH et al. (2016), which found that impaired LA reservoir strain was a stronger predictor of mortality among patients with HFpEF than LV or RV strain measures [9]. The superior predictive value of LA functional parameters compared to LVDD parameters may be related to the essential role of a disproportionate (intrinsic) atrial myopathy in the pathophysiology of HFpEF. In disproportionate LA myopathy, or ‘stiff atrial syndrome’, LA dysfunction develops out of proportion to LVDD and usually progresses faster than LVDD [31]. Due to its high prevalence in HFpEF [32], disproportionate LA myopathy as assessed by functional LA parameters may eventually describe better prognosis than measures of LVDD as well as LV hypertrophy. In arterial hypertension, LV hypertrophy develops to compensate for high LV afterload, and the necessary balance is usually achieved during the asymptomatic stage [33]. At the symptomatic, or HFpEF stage, LV mass virtually ceases to increase [33], which may partially explain the lack of association of LVMI with adverse outcomes in the present study, despite LVH being present in most participants.

The deleterious prognostic impact of LA dysfunction in HFpEF can be mediated through the development of AF [31], which is an important example of disproportionate LA myopathy [34]. Although we did not include patients with permanent AF, paroxysmal type of AF was observed in a large proportion (45%) of participants. In the present study, paroxysmal AF was associated with structural and functional LA deterioration, which is in line with the findings of Reddy YNV et al. (2020), where LA mechanical properties progressively decline with increasing AF burden in HFpEF [35]. In the present study, paroxysmal AF also heralds an adverse outcome among hypertensive patients with HFpEF, which is also consistent with previous studies [36,37]. Given the pathophysiological similarity and reciprocal influence of LV dysfunction and AF (the more severe the LV dysfunction, the higher the risk of AF, and vice versa), the similar prognostic value for paroxysmal AF and LASr shown in this study appears to be quite reasonable and largely expected.

The high prognostic role of AF could be explained by the fact that onset of AF during the natural course of HFpEF signifies quite pronounced LA impairments [6,31] mainly due to atrial fibrosis. LA fibrosis, in turn, reduces atrial distensibility with increased LA pressure during the reservoir phase and reactive pulmonary component involvement (increased pulmonary vascular resistance), which places an increased load on the RV [35] with secondary development of RV dysfunction [38,39]. Of note, in the TOCPAT (Treatment of Preserved Cardiac Function Heart Failure with an Aldosterone Antagonist) trial, LA functional parameters were more important risk markers for the occurrence of AF than LA enlargement [26].

The independent prognostic impact of paroxysmal AF that we have identified may be of great importance for clinical practice, as it may indicate the need for aggressive ablation strategies at this earlier stage of AF, when atrial fibrosis has not yet reached its maximum and there is a high probability of significant LA reduction and decreased adverse event likelihood. In advanced stages of AF, there is a high risk of dyspnoea exacerbation after ablation due to the addition of ablation-related injury to pre-existing severe diffuse atrial fibrosis [31].

Thus, our data provides valuable insights into the prognostic significance of LA dysfunction in HFpEF, and the development of treatment strategies to maintain LA function is of paramount importance. Several devices that decompress the LA, such as transcatheter implantation of an interatrial shunt to redirect blood flow to the right atrium, have produced promising haemodynamic and prognostic results [40]. Therapeutic interventions aimed at reverse structural and functional remodeling of the LA may also be important. In a recent prospective study in patients with HFpEF and T2DM, the use of the SGLT2 inhibitor empagliflozin for 6 months was associated with a significant reduction in LA volume, improvement in LA reservoir strain, and restoration of atrial reserve compared to the control (an enhancement of LASr increase during exercise) [41]. In severe LA myopathy, a three-component antifibrotic/anti-inflammatory therapy consisting of an SGLT2 inhibitor, an ARNI and a mineralocorticoid receptor antagonist may have certain potential. This concept is currently being tested in an ongoing prospective clinical study in the Russian Federation (clinicaltrials.gov registration number NCT06655480).

Some limitations to this study should be noted. First, we only used LASr of all the available indices of LA mechanics, which could lead to certain discrepancies. Nevertheless, LA reservoir strain remains the most carefully studied parameter of LA deformation, highly reproducible and feasible and one of the most powerful diagnostic and prognostic parameters in HFpEF [9]. Second, LA deformation was analyzed only in the apical four-chamber view, whereas most other studies used biplane LA longitudinal strain. However, according to the current consensus [19], it is acceptable to rely only on LA longitudinal strain obtained from a non-foreshortened apical four-chamber view. Third, the predictive performance of LA functional parameters was only moderate, which may limit the clinical significance of this finding, particularly with regard to decision-making. However, the moderate significance of LASr appears to be quite typical for HFpEF, given the extraordinary heterogeneity of this disease’s phenotypic manifestations, where no single indicator can perfectly reflect the severity of the disease or predict the prognosis [42,43]. At the same time, the present study showed that LASr of less than 18% had a high specificity of 92% for identifying patients with adverse outcomes. This threshold is used in the updated ASE algorithm for assessing LV diastolic function/filling pressure [16] and, therefore, can be applied for reasonably accurate assessing an unfavorable prognosis. Fourth, our study excluded subjects with permanent AF, which may have limited our ability to determine the true prognostic significance of AF burden in HFpEF patients. Nevertheless, the identification of the powerful prognostic significance of paroxysmal AF in the present study clearly confirms the crucial role of AF as a marker of HFpEF progression. Fifth, as most patients were enrolled in studies prior to the introduction of modern HFpEF drugs, relatively few patients received ARNIs, SGLT2 inhibitors or mineralocorticoid receptor antagonists, which restrict the generalizability of the findings. Finally, unlike in most recent large-scale trials, we did not consider elevated NT-proBNP levels to be an inclusion criterion. At the same time, a large proportion of patients with invasively proven HFpEF have normal NT-proBNP levels, predominantly obese patients with concentric LVH [44], and an elevated NT-proBNP level is no longer a mandatory diagnostic criterion for HFpEF [45].

## 5. Conclusions

In the present retrospective cohort study, an LA functional parameter reservoir strain seems to be more effective than parameter of LA structure (maximal volume index) or traditional measures of LV diastolic function/filling pressure (E/e′ ratio and PASP) in predicting long-term prognosis in a well-defined cohort of ambulatory hypertensive patients with stable HFpEF. Another powerful marker of adverse outcomes was the presence of paroxysmal AF, which has intrinsic proximity with LA dysfunction. Further studies are needed to determine whether strategies aimed at improving LA function or preventing/treatment of paroxysmal AF could contribute to slowing the disease progression and ultimately be beneficial in HFpEF. 

## Figures and Tables

**Figure 1 life-15-01483-f001:**
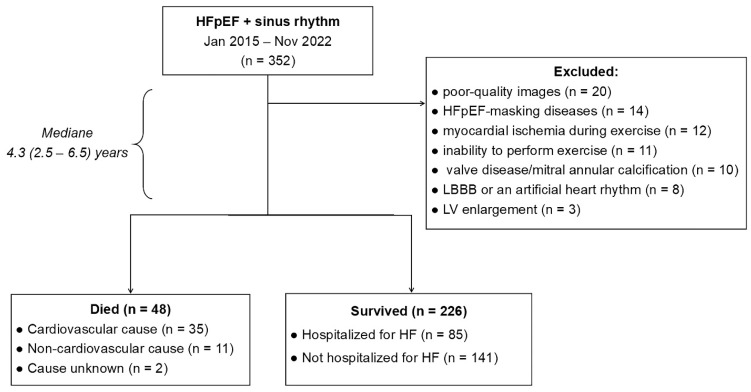
Flow chart of patient enrolment. HFpEF, heart failure with preserved ejection fraction; LBBB, left bundle branch block; LV, left ventricular.

**Figure 2 life-15-01483-f002:**
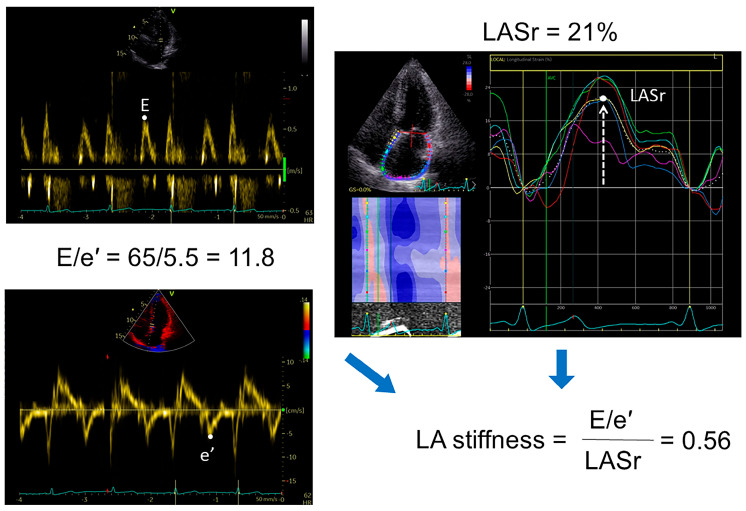
Example of calculating the mitral early diastolic-to-annulus relaxation velocity (E/e′) ratio, total longitudinal strain of the left atrium in the reservoir phase (LASr), and left atrial stiffness. LA strain was determined as the average value of the longitudinal positive strain peak during LA relaxation from all six segments of the LA in the apical 4-chamber view, using the onset of the QRS as the referent point. The white dotted curve represents the average value of LA strain from all analysed LA segments; the colour curves represent strains of separate segments.

**Figure 3 life-15-01483-f003:**
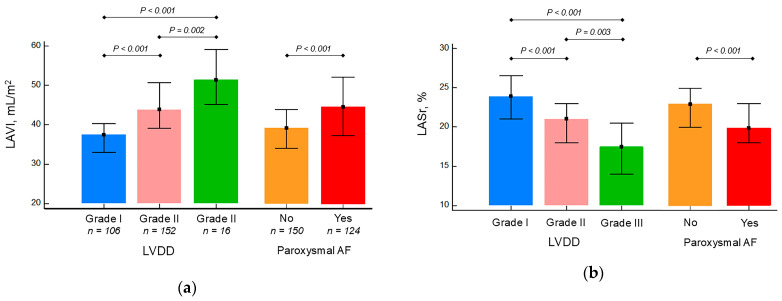
Comparisons of left atrial echocardiographic parameters (LAVI (**a**), LASr (**b**), and LA stiffness (**c**)) in hypertensive patients with HFpEF and various degrees of LV diastolic dysfunction or status of AF; correlation between left atrial structural (LAVI) and functional (LASr and LA stiffness) parameters (**d**,**e**). The bars indicate the median, and the markers indicate the interquartile ranges. AF, atrial fibrillation; HFpEF, heart failure with preserved ejection fraction; LASr, left atrial strain during the reservoir phase; LAVI, left atrial volume index; LVDD, left ventricular diastolic dysfunction.

**Figure 4 life-15-01483-f004:**
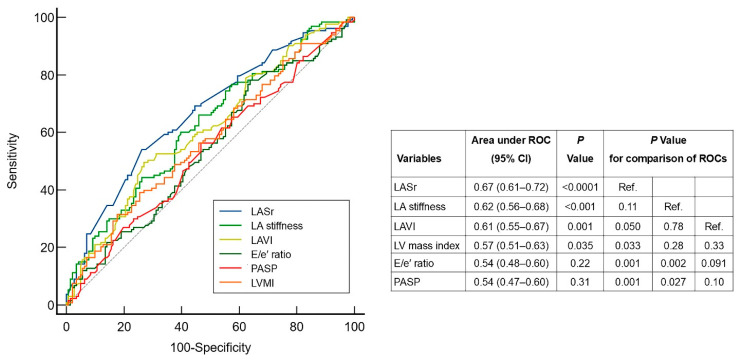
Comparison of the prognostic performance among echocardiographic parameters reflecting the LA structure/function and LV diastolic dysfunction/filling pressure for long-term adverse outcomes. E, early inflow velocity; e′, averaged annulus relaxation velocity; LA, left atrial; LASr, left atrial strain during the reservoir phase; LAVI, left atrial volume index; LV, left ventricular; PASP, pulmonary artery systolic pressure.

**Figure 5 life-15-01483-f005:**
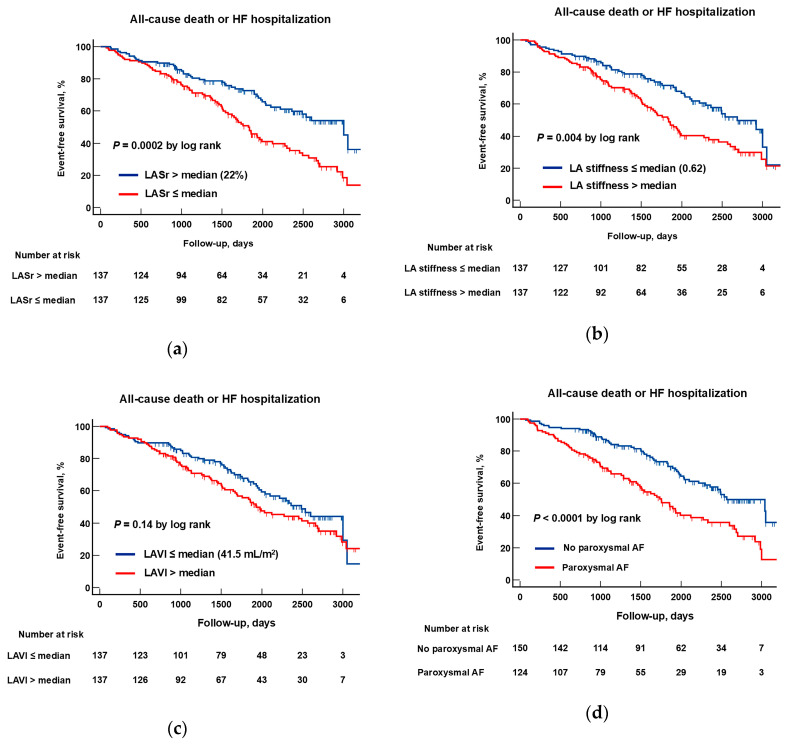
Kaplan–Meier survival curves showing time to death from any cause or hospitalization for HF in patients with HFpEF, stratified according to baseline structural and functional left atrial parameters (LASr (**a**), LA stiffness (**b**), LAVI (**c**)); paroxysmal atrial fibrillation (AF, (**d**)), and LV diastolic dysfunction/filling pressure (E/e′ ratio (**e**), PASP (**f**), and LVMI (**g**)). Median cutoff values (≤median vs. >median) were used for the analysis. The censored data are marked in the graph with a small vertical line. E, early mitral inflow velocity; e′, averaged annulus relaxation velocity; LASr, left atrial strain during the reservoir phase; LVMI, left ventricular myocardial index; LAVI, left atrial volume index; PASP, pulmonary artery systolic pressure.

**Table 1 life-15-01483-t001:** Baseline clinical and biochemical variables of patients with HFpEF depending on the outcome of the disease.

Variables	All Patients (*n* = 274)	Died or Hospitalized for HF (*n* = 133)	Survived Without HF Hospitalization (*n* = 141)	*p* Value
*Clinical variables*				
Age, y	68.7 (63.6–77.0)	69.7 (64.1–76.8)	67.9 (61.7–74.0)	0.031
Male gender, n (%)	121 (44)	58 (44)	63 (45)	0.86
Body mass index, kg/m^2^	32.0 ± 5.4	31.3 ± 5.1	32.7 ± 5.6	0.037
Systolic BP, mm Hg	140 (130–150)	140 (130–150)	138 (130–145)	0.13
Diastolic BP, mm Hg	85 (80–90)	86 (80–93)	85 (80–90)	0.73
Heart rate, min^−1^	65 (60–71)	65 (60–71)	65 (61–71)	0.95
6-MWTD, m	329 ± 77	322 ± 76	335 ± 78	0.16
NYHA functional class:				0.10
I, n (%)	41 (15)	14 (11)	27 (19)	0.046
II, n (%)	160 (58)	79 (59)	81 (57)	0.74
III, n (%)	73 (27)	40 (30)	33 (23)	0.21
NT-proBNP, pg/mL	283 (193–449)	312 (223–630)	253 (158–365)	<0.0001
eGFR, mL/min/1.73 m^2^	64 (46–72)	63 (44–72)	65 (50–74)	0.10
*Comorbidities and associated conditions*				
Overweight/obesity, ^1^ n (%)	250 (91)	121 (91)	129 (92)	0.88
Hypertension, ^2^ n (%)	274 (100)	133 (100)	141 (100)	0.30
Paroxysmal atrial fibrillation, n (%)	124 (45)	76 (57)	48 (34)	0.0001
Ischemic heart disease, n (%)	96 (35)	48 (36)	48 (34)	0.72
Previous myocardial infarction, n (%)	40 (15)	20 (15)	20 (14)	0.84
Myocardial revascularization, n (%)	74 (27)	36 (27)	28 (20)	0.16
Type 2 diabetes mellitus, n (%)	110 (40)	54 (41)	56 (40)	0.88
Chronic kidney disease, ^3^ n (%)	112 (41)	60 (45)	52 (37)	0.17
Stroke, n (%)	15 (5)	10 (8)	5 (4)	0.15
Anemia, n (%)	36 (13)	21 (16)	15 (11)	0.21
COPD, n (%)	45 (16)	23 (17)	22 (16)	0.51
*Cardiovascular medications*				
ACEI/ARB, n (%)	220 (80)	112 (84)	108 (77)	0.11
ARNI, n (%)	49 (18)	20 (15)	29 (21)	0.23
SGLT2 inhibitors, n (%)	39 (14)	19 (14)	20 (14)	0.98
MRA, n (%)	36 (13)	13 (10)	23 (16)	0.11
Calcium channel blocker, n (%)	125 (46)	58 (44)	67 (48)	0.52
β-Blockers, n (%)	186 (68)	90 (68)	96 (68)	0.94
Loop diuretics, n (%)	203 (74)	108 (81)	95 (67)	0.009
Statins, n (%)	200 (73)	97 (73)	103 (73)	0.98

Data are presented as the means ± standard deviations for continuous normally distributed variables, the medians (25th–75th percentiles) for nonnormally distributed continuous variables, and frequencies (%) for categorical variables. ^1^—body mass index ≥ 30 kg/m^2^; ^2^—blood pressure ≥ 140/90 Hg mm; ^3^—eGFR < 60 mL/min/1.73 m^2^. ACEI, angiotensin-converting enzyme inhibitor; ARB, angiotensin receptor blocker; ARNI, angiotensin receptor–neprilysin inhibitors; COPD, chronic obstructive pulmonary disease; eGFR, estimated glomerular filtration rate; HFpEF, heart failure with preserved ejection fraction; MRA, mineralocorticoid receptor antagonist; NT-proBNP, N-terminal pro-brain natriuretic peptide; NYHA, New York Heart Association; SGLT2, sodium–glucose cotransporter 2; 6-MWTD, 6 min walk test distance.

**Table 2 life-15-01483-t002:** Baseline echocardiographic variables of patients with HFpEF depending on the outcome of the disease.

Variables	All Patients (*n* = 274)	Died or Hospitalized for HF (*n* = 133)	Survived Without HF Hospitalization (*n* = 141)	*p* Value
LV ejection fraction, %	59.8 (55.8–65.5)	59.4 (56.3–66.7)	59.8 (55.4–65.2)	0.45
LV GLS, %	18.5 ± 3.2	18.7 ± 2.9	18.2 ± 3.4	0.42
LV EDV, mL	93 (78–110)	92 (80–108)	94 (75–111)	0.98
LV mass index, g/m^2^	110.6 (101.1–127.7)	112.9 (102.4–131.7)	109.0 (99.1–122.5)	0.037
LV hypertrophy, ^1^ n (%)	188 (69)	97 (73)	91 (65)	0.14
Relative wall thickness	0.50 ± 0.09	0.50 ± 0.08	0.50 ± 0.09	0.98
*Types of LV geometry*				0.35
Normal geometry	21 (8)	10 (8)	11 (8)	0.93
Concentric remodeling	65 (24)	26 (19)	39 (28)	0.12
Concentric hypertrophy	165 (60)	87 (65)	78 (55)	0.089
Eccentric hypertrophy	23 (8)	10 (8)	13 (9)	0.61
LAVI, mL/m^2^	41.5 (36.7–49.0)	43.4 (37.6–50.0)	40.0 (35.5–44.9)	0.002
LA dilatation, ^2^ n (%)	230 (84)	120 (90)	110 (78)	0.006
LASr, %	22 (19–24)	20 (18–23)	23 (20–26)	<0.0001
LA dysfunction, ^3^ n (%)	159 (58)	93 (70)	66 (47)	0.0001
LA stiffness	0.62 (0.45–0.80)	0.67 (0.51–0.90)	0.57 (0.43–0.74)	0.0006
E/A ratio	0.88 (0.70–1.16)	0.83 (0.68–1.25)	0.90 (0.76–1.13)	0.27
E/e′ ratio	13.6 (10.8–15.8)	13.7 (11.3–16.2)	13.2 (10.3–15.8)	0.22
E/e′ > 14, n (%)	123 (46)	63 (47)	60 (43)	0.42
*LV diastolic dysfunction:*				0.014
I, n (%)	106 (39)	42 (32)	64 (45)	0.019
II, n (%)	152 (55)	79 (59)	73 (52)	0.21
III, n (%)	16 (6)	12 (9)	4 (3)	0.029
PASP, mm Hg	35.0 (28.0–43.1)	36.9 (28.8–46.9)	33.3 (27.0–40.9)	0.025
Pulmonary hypertension, ^4^ n (%)	138 (50)	72 (54)	66 (47)	0.23
TAPSE, cm	2.1 ± 0.4	2.1 ± 0.4	2.2 ± 0.4	0.11
RV dysfunction, ^5^ n (%)	43 (16)	23 (17)	20 (14)	0.48
Increased RA pressure, ^6^ n (%)	129 (47)	64 (48)	65 (46)	0.74

Data are presented as the means ± standard deviations for continuous normally distributed variables, the medians (25th–75th percentiles) for nonnormally distributed continuous variables, and frequencies (%) for categorical variables. ^1^—LV mass index > 115 g/m^2^ in men and >95 g/m^2^ in women; ^2^—LA volume index ≥ 34 mL/m^2^; ^3^—LASr < 23%; ^4^—PASP > 35 mm Hg; ^5^—TAPSE ≤ 1.7 cm; ^6^—RA pressure > 3 mm Hg. A, late inflow velocity; E, early inflow velocity; e′, averaged annulus relaxation velocity; EDV, end-diastolic volume; EF, ejection fraction; GLS, global longitudinal strain; LA, left atrial; LASr, left atrial strain during the reservoir phase; LAVI, left atrial volume index; LV, left ventricular; PASP, pulmonary artery systolic pressure; RA, right atrial; TAPSE, tricuspid annular plane systolic excursion.

**Table 3 life-15-01483-t003:** Predictors of an adverse outcome (all-cause mortality or hospitalization due to HF exacerbation) during follow-up.

Parameters	Univariable		Multivariable					
	Unadjusted HR (95% CI)	*p* Value	Unadjusted HR (95% CI)—Model 1	*p* Value	Adjusted HR (95% CI) ^1^—Model 2	*p* Value	Adjusted HR (95% CI) ^2^—Model 3	*p* Value
LASr	0.93 (0.90–0.97)	0.0001	0.95 (0.90–0.99)	0.025	0.95 (0.90–0.99)	0.023	0.94 (0.90–0.99)	0.023
Paroxysmal AF	2.03 (1.43–2.87)	0.0001	1.71 (1.17–2.51)	0.006	1.71 (1.16–2.50)	0.006	1.77 (1.19–2.64)	0.005
LA stiffness	1.75 (1.13–2.71)	0.012	–	–	–	–	–	–
No loop diuretics	0.57 (0.37–0.89)	0.012	0.79 (0.47–1.31)	0.36	0.81 (0.48–1.37)	0.43	0.86 (0.51–1.45)	0.58
LAVI	1.02 (1.01–1.04)	0.013	0.99 (0.97–1.02)	0.56	0.99 (0.97–1.02)	0.71	1.00 (0.97–1.03)	0.83
NYHA fc	1.35 (1.05–1.75)	0.020	1.15 (0.86–1.53)	0.36	1.06 (0.81–1.47)	0.58	1.10 (0.80–1.52)	0.55
LVDD	1.35 (1.01–1.79)	0.042	0.91 (0.63–1.32)	0.63	0.94 (0.64–1.36)	0.73	0.97 (0.66–1.42)	0.87
Systolic BP	1.01 (1.00–1.02)	0.073	–	–	–	–	1.01 (1.00–1.03)	0.058
Age	1.02 (1.00–1.04)	0.097	–	–	1.01 (0.99–1.03)	0.48	1.01 (0.99–1.03)	0.42
Male gender	0.78 (0.55–1.10)	0.15	–	–	0.91 (0.63–1.33)	0.63	0.90 (0.61–1.33)	0.60
NT-proBNP	1.01 (0.99–1.09)	0.18	–	–	–	–	1.00 (0.99–1.01)	0.27
CKD	1.23 (0.87–1.73)	0.25	–	–	–	–	1.03 (0.70–1.51)	0.88
T2DM	1.16 (0.82–1.66)	0.40	–	–	–	–	1.03 (0.69–1.53)	0.89
LVMI	1.00 (0.99–1.01)	0.41	–	–	–	–	1.00 (0.99–1.01)	0.56
PASP	1.00 (0.98–1.01)	0.51	–	–	–	–	–	–
E/e′ ratio	1.01 (0.97–1.06)	0.52	–	–	–	–	–	–
BMI	1.00 (0.97–1.04)	0.88	–	–	1.02 (0.98–1.06)	0.34	1.02 (0.98–1.06)	0.35

^1^ Adjusted for age, sex, and body weight index; ^2^ Adjusted for age, sex, body weight index, NT-proBNP, systolic BP, LV mass index, chronic kidney disease, diabetes mellitus. AF, atrial fibrillation; CI, confidence interval; CKD, chronic kidney disease; fc, functional class; HR, hazard ratio; LVDD, left ventricular diastolic dysfunction; T2DM, type 2 diabetes mellitus. The remaining abbreviations are the same as in Table 1 and Table 2.

## Data Availability

The authors confirm that the data supporting the findings of this study are available within the article. Raw data that support the findings of this study are available from the corresponding author, upon reasonable request.

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
