# Peer review of "The Prognostic Role of the Left Atrium in Hypertensive Patients with HFpEF: Does Function Matter More than Structure?"

_life, 2025, doi:10.3390/life15091483_

Round 1

Reviewer 1 Report

Comments and Suggestions for Authors

The paper is interesting, but several points need to be clarified

Due to the wide overlap of 25th-75th percentile ranges (why not using means +- SD ?) the conclusions of the Abstract and of the Paper should be attenuated in strenght. In fact the observed results can be accepted only at the population level, while examining the individual cases each value is very difficult to be interpreted, due to overlap of 25th-75th percentiles ranges and the very close absolute values (3% difference) of medians between the two groups (please check the spelling of the term “median” versus “mediane” throughout the paper and in Figure 1, in lines 324-325 and so on). This means that LASr can not be considered useful, at the present time, in clinical practice in the decision making process in the individual patient. This should be clearly stated in the Discussion and Conclusions. 

Why presenting the data in Table 1 and Table 2 as medians and 25th-75Th percentile ranges?  To better compare the results  with other studies, why data could not be presented as means +- SD ? The same for Figure 1 and 3. Furthermore Figure 3 is clearly showing wide overlap for each parameter amog different groups. In Figure 3, in quadrants d and e, the dispersion along the lines is even greater, with a correlation coefficient  statistically significant, but apparently not useful for clinical decision making in individual subjects. All this should be discussed in the Discussion and Limitations, and taken into account in the Conclusions.

Lines 91-92: it is not clear how many patients had coronary angiograhy done and with which indications? Was complete myocardial revascularization performed in how many patients? Or was only target vessel revascularization performed ?

Line 111: were the 352 all consecutive patients ? It is not clear when, during the time course of the disease, all the measurement were performed ? How baseline was defined in this time course ? were all outpatients ? how many patients had previous admissions to the hospital ? and for which reasons? 

Lines 115-117: since in this population almost 50% met an end-point, this is a very high risk population where you already know from the beginning that one out of two cases will have complications. All this is reducing the value of any further prognostic analysis, since you already know that you should do a very active interventional strategy. How interventional strategies were applied and how those interventions could have changed the prognosis ?Also this should be discussed in the discussion and limitations. In fact in Table 1 there is a strong prevalence of obesity, paroxysmal AF, ischemic heart disease, diabetes  and chronic kidney disease (how this was defined since eGFR values are not so low ?).

Lines 145-146 a clear definition of different classes of LVDD should be reported.

Table 1: how the diagnoses of HFpEF was reached in patients with NYHA class I and with NT-proBNP values generally low ?

Lines 174-186: It is not clear how the results of this test were used in the present study.

Lines 290-309: the predictive performance of all the variables appear statistically significant but not very useful from a clinical point of view, beeing around 50%. For the best parameter (lines 295-296 and Figure 4) it is said: “The prognostic performance of LASr was highest (cutoff value 20%) with an AUC 0.67 (95% CI 0.61 to 0.72), P < 0.0001.” This means that aronud 30% of cases can not be correctly classified. This also seem clinically not so usable nor useful.

For paragraph 3.4. Identification of prognostic markers for HFpEF : lines 310-352 and Figure 5. It si not clear how the cut off value of 20% median could be clinicaly useful in individual patients when difference between medians of the two groups is 3%, and 25th-75% percentile ranges are widely overlapping? To evaluate the feasibility and clinical applicabiliy of these data a reproducibility of the measurements with interobserver and intraobserver variability should be reported.

Table 3 is showing that the strenght of prediction of adverse outcome is very small for LASr, even though statistically significant. The clinical prediction capability of paroxysmal AF is a lot stronger (almost twice) and clinically more useful.

The Discussion appears too long and repetitive. The Discussion should be then shortened, taken also into account all the limitations and all the above considerations.

Also the conclusions should be modified accordingly, taking also into account that the predictive value of paroxysmal AF was stronger than the echocardiographic parameter LASr.    

Author Response

Please, see attachment

Reviewer 2 Report

Comments and Suggestions for Authors

This research, conducted by Ovchinnikov et al., presents substantial evidence that the functional assessment of the left atrium, namely reservoir strain (LASr), is more effective than structural indices like LAVI in forecasting long-term outcomes in hypertensive patients with HFpEF. The authors utilized a comprehensive echocardiographic technique and a thoroughly defined cohort with a median follow-up duration surpassing four years, hence enhancing the validity of their results. Significantly, LASr and paroxysmal atrial fibrillation were identified as independent prognostic indicators, in contrast to traditional assessments of diastolic dysfunction or hypertrophy, which were not.

The retrospective approach and the pre-modern therapeutic environment (limited exposure to SGLT2 inhibitors/ARNI) restrict the generalizability of the findings. Nevertheless, this study underscores left atrial (LA) dysfunction as a critical factor in the evolution of heart failure with preserved ejection fraction (HFpEF) and advocates for the integration of LA functional MRI into standard clinical practice. It is an important addition to the expanding body of evidence that functional rather than structural atrial assessment should be used to determine who is at risk in this group of people who are at high risk.

Reviewer 3 Report

Comments and Suggestions for Authors

The subject of the article is very interesting and also with practical aplications.

Starting from the ideea that LA dysfunction may be a strong prognostic predictor than structural parameters in patients with HfpEF, the aim  of the study was to evaluate the long-term prognostic significance of LA structural vs. functional parameters and to compare the prognostic impact of LA parameters to that of parameters reflecting LV diastolic function in hypertensive patients with HFpEF.

The design ofthe study is well conceived, clear. The evaluation of LA may be one of the key points in patients with HfpEF. I suggest that in the introduction some ideas be added about the particularities of the structural and functional changes of the left atrium strictly related to arterial hypertension. Only general things are presented.

What about the relation between LA changes and the risk of atrial fibrillation? More details are needed because AF is one ofthe most frequent complications of AH.

Some of my suggestions are inserted directly into the text

Some references are old and very old and one maybe two with no direct relatin with the subject. I have marked them with colour. I suggest to replace them

Comments on the Quality of English Language

minor english corrections are needed

Round 2

Reviewer 1 Report

Comments and Suggestions for Authors

The Authors made the required changes.